# A Rapid Method for Measuring In Vitro Growth in Entomopathogenic Fungi

**DOI:** 10.3390/insects14080703

**Published:** 2023-08-13

**Authors:** Anna R. Slowik, Helen Hesketh, Steven M. Sait, Henrik H. de Fine Licht

**Affiliations:** 1Department of Plant and Environmental Sciences, University of Copenhagen, Thorvaldsensvej 40, 1871 Frederiksberg, Denmark; 2UK Centre for Ecology & Hydrology, Maclean Building, Benson Lane, Crowmarsh Gifford, Wallingford OX10 8BB, UK; hhesketh@ceh.ac.uk; 3School of Biology, Faculty of Biological Sciences, University of Leeds, Leeds LS2 9JT, UK; s.m.sait@leeds.ac.uk

**Keywords:** *Metarhizium*, fungal growth, filamentous fungi, bioassay technique, spectrophotometry, microspectrophotometry, microplate reader, biomass quantification

## Abstract

**Simple Summary:**

This study aimed to develop a rapid and efficient method for measuring the growth of entomopathogenic fungi, which are used as environmentally friendly alternatives to chemical insecticides. The traditional methods used to measure fungal growth are time-consuming and limited in their ability to capture important aspects of growth. In this research, we implemented an indirect measure using a microplate reader, which uses the optical density of small-volume cultures to estimate fungal growth. We directly related changes in optical density to the amount of fungal biomass, and compared the results of this method with traditional measurements on solid agar plates. We found that the microspectrophotometric approach provided accurate and reliable measurements. Our findings revealed differences in growth rates and biomass production among different species and isolates of the fungi. This technique offers a valuable tool for studying the growth dynamics of entomopathogenic fungi, and has practical applications in assessing their virulence and pathogenic potential. It can contribute to our understanding of how these fungi grow during infection, and aid in the development of effective biological control strategies against insect pests.

**Abstract:**

Quantifying the growth of entomopathogenic fungi is crucial for understanding their virulence and pathogenic potential. Traditional methods for determining growth, such as biomass determination or colony growth area, are time-consuming and quantitatively and spatially limited in scope. In this study, we introduce a high-throughput method for rapidly measuring fungal growth using spectrophotometry in small-volume, liquid media cultures in 96-well microplates. Optical density (OD) changes were directly correlated with dry weight of samples for six isolates from three species of the genus *Metarhizium* to validate spectrophotometric growth measurements, and investigate species- and isolate-specific effects. We quantified fungal biomass from the microcultures by extracting, drying, and weighing mycelial mats. From the relationship established between OD and biomass, we generated standard curves for predicting biomass based on the OD values. The OD measurements clearly distinguished growth patterns among six isolates from three *Metarhizium* species. The logistic growth phase, as captured by the OD measurements, could be accurately assessed within a span of 80 h. Using isolates of *M. acridum*, *M. brunneum*, and *M. guizhouense*, this technique was demonstrated to be an effective, reproducible, and simple method for rapidly measuring filamentous fungal growth with high precision. This technique offers a valuable tool for studying the growth dynamics of entomopathogenic fungi and investigating the factors that influence their growth.

## 1. Introduction

Entomopathogenic fungi are important natural regulators of insect populations, and are widely used as environmentally friendly biological control alternatives to synthetic chemical insecticides [1,2,3,4]. In biological studies, measurements of fungal growth over time on different defined media serve as a standard measure to assess performance and evaluate efficacy during isolation and testing [5,6]. Being heterotrophic organisms with indeterminate growth, fungal growth often responds directly to the quality and availability of nutrients in the immediate environment [7,8]. Fungal growth over time can be evaluated using various methods, generally classified as direct or indirect measures [9]. The most widely applied direct methods include measurement of hyphal extension [10], dry weight from liquid cultures, and radial expansion on solid agar [11,12]. Accepted indirect methods make use of spectrophotometry in liquid macro- or micro-cultures [10,13,14], fluorescence of labeled fungi and light sheet fluorescence [15,16], multispectral imaging [17], quantification of chitin production [18,19], and advances in the spectrophotometric analysis of microcultures on agar media [20].

Measurement of the radial expansion of fungal colonies on solid media is a widely-used direct method to quantify growth. The method is straightforward, and allows for measurement of other phenotypic traits such as spore production and colony morphology (e.g., color, branching pattern) [6,8,21]. However, this approach fails to account for some important aspects of growth, such as the density of the mycelium [20]. An expanding colony can exhibit varying degrees of mycelial density while covering the same area, which is not captured when using radial expansion measurements. Using a dry weight method to measure growth in liquid macrocultures accounts for this discrepancy in mycelial density, but requires interference with cultures through direct sampling for quantification, making real-time monitoring of in situ growth impractical [13,22].

In addition to this, the solid media environment may not be biologically appropriate for some fungi based on their specific ecologies. In entomopathogenic fungi, in vivo growth progresses primarily through the insect hemocoel, which is a submerged liquid environment [23]. The physiological and developmental biology of fungi can vary considerably, depending on whether they grow on a solid or liquid medium. For instance, solid media may support a higher production of secondary metabolites or enzymes compared to liquid media in some fungal isolates [24,25]. In terms of practicality, fungal growth on agar plates is also time-consuming to conduct. The timescale for growth of many entomopathogenic fungi when analyzing radial expansion spans multiple days, and in the case of measuring dry weight, the processing of samples involves a lengthy process of collection, drying, and weighing [3,9,21].

In situ spectrophotometry provides a viable alternative to these methods by directly correlating optical density (OD) values with an increase in fungal biomass. It is generally understood that growing fungus changes the OD of liquid cultures, because turbidity directly correlates with unit population size, which serves as the basis for the traditional spectrophotometric analysis of fungal growth [26]. In this study, we apply spectrophotometric analysis to measure the growth of entomopathogenic fungi, which are typically performed on solid agar media, as previously described. Previous studies have demonstrated the use of spectrophotometric measurements for assessing filamentous fungal growth in microcultures. However, in these studies, the relationship between dry mycelial mass and OD is extrapolated using correlation coefficients [13], or hyphal extension is employed as a growth metric [10]. The aim of this study was to establish a direct correlation between the dry weight of mycelial cultures and their corresponding OD values for six isolates of *Metarhizium* spp. This correlation allows for the construction of isolate-specific standard curves, enabling the quantification of biomass based on OD measurements.

## 2. Materials and Methods

### 2.1. Fungal Isolates and Preparation of Inoculum

The growth of two isolates within each of three different species of *Metarhizium* were compared to investigate variations among species and isolates. This was assessed in addition to the effect on the relationship between dry weight and OD. Six fungal isolates of the genus *Metarhizium* were used to produce standard curves of OD by dry weight: *M. brunneum* KVL 16_36 (Isolated from the commercial product Met52, Novozymes A/S, Krogshøjvej 36, Bagsværd, Denmark), *M. brunneum* KVL 12_30 [27], *M. acridum* KVL 18_06 (ARSEF 6421), *M. acridum* KVL 04_55 (ARSEF 7486), *M. guizhouense* KVL 19_24 (ARSEF 977), and *M guizhouense* KVL 19_28 (ARSEF 3611). The acronym ARSEF refers to the United States Department of Agriculture (USDA) Agricultural Research Service (ARS) collection of Entomopathogenic Fungal cultures (https://data.nal.usda.gov/dataset/ars-collection-entomopathogenic-fungal-cultures-arsef. URL accessed on 11 March 2022). The acronym KVL refers to the entomopathogenic fungus culture collection maintained at the Section for Organismal Biology, Department of Plant and Environmental Sciences, University of Copenhagen. *Metarhizium* (Metschnikoff) Sorokin (Order Hypocreales: Family Clavicipitaceae) was selected as the focus of our study, due to its multifaceted importance in both evolutionary ecology and practical applications in pest management [2,3,4].

The fungal cultures were grown on quarter-strength Sabouraud dextrose agar + yeast media (SDAY/4: 2.5 g/L 1:1 animal:bacterial peptone (bacteriological peptone and Acuferm Neoeptone, Neogen Corp., 620 Lesher Place, Lansing, MI, USA), 10 g/L dextrose (Bacteriological, Oxoid Ltd., Wade Road, Hampshire, UK), 2.5 g/L yeast extract (Neogen Corp.), 15 g/L agar (Bacteriological (European Type) No. 1, Neogen Corp.))) in Petri dishes (90 mm × 15 mm triple-vented, Sterilin Ltd., 1 Ashley Road, Altrincham, Cheshire, UK) at 23 °C, and conidia were harvested after 14 days. The conidia were collected in 0.1% (*v*/*v*) Tween^®^ 80 (Merk KgaA, Frankfurter Straße 250, Darmstadt, Germany) via agitation with a Drigalski spatula from sporulating colonies, and the resulting suspension was centrifuged; the supernatant was removed, and the colonies were rinsed twice with 0.1% (*v*/*v*) Tween^®^ 80 to remove all fragments of mycelia. Conidial suspensions were prepared at a concentration of 2 × 10^6^ conidia per mL by dilution using 0.1% (*v*/*v*) Tween^®^ 80. The concentration of the stock suspension was determined by counting conidia from 1000× serially diluted stock suspension in a Fuchs Rosenthall hemocytometer (×400 magnification). To verify germination, 100 µL of 100× diluted stock suspension was spread with a Drigalski spatula on an agar plate of SDAY/4 and incubated for 24 h at 23 °C. Four microscope coverslips (22 mm × 22 mm) were then placed over the culture surface, and 100 conidia were counted under each coverslip. Conidia were considered to have germinated with the germ tube that was at least as long as the width of the conidium, and conidial germination after 24 h was verified as >98% in all cases before being used in further assays.

### 2.2. Continuous Growth Curve

To determine whether measurement by spectrophotometry in liquid microcultures can accurately capture the growth curve of an entomopathogenic fungus, a growth curve of *M. brunneum* KVL 12_30 was produced. The microcultures were prepared in 96 wells of a clear flat-bottom vented microplate (Starlab International GmbH, Neuer Höltigbaum 38, Hamburg, Germany) by inoculating 100 µL of 2 × 10^6^ conidial suspension into 100 µL Sabouraud dextrose + yeast media (SDY/4: 2.5 g/L 1:1 animal:bacterial peptone, 10 g/L dextrose, 2.5 g/L yeast extract). The OD of each well was measured at 405 nm in a Synergy^TM^ HT MultiDetection Microplate Reader (BioTek Instruments Ltd., Cheadle, UK) with Gen5 software Version 2.00.18 every 10 min for 96 h at 24 °C, without removal of the microplate from the plate reader. The wavelength of 405 nm was previously described as being fit for this purpose [10]. Pilot assays were performed comparing a range of wavelengths for measuring growth in microcultures that confirmed this. The microcultures were checked visually for bacteria using a compound microscope upon completion to ensure they were uncontaminated.

### 2.3. Standard Curve for Fungal Dry Weight and OD

To establish a correlation between OD and fungal biomass, OD values were measured, and the mycelial mats were subsequently extracted and weighed for the six isolates at four time points (20, 40, 60, and 80 h post-inoculation) during the linear growth phase. The period of logistic growth was determined from the continuous growth curve generated, as detailed in Section 2.2, utilizing the described parameters for media and conidial suspension. The fungal microcultures were prepared as described in Section 2.2. For each measurement at the four time points, a single microplate containing 60 wells was utilized to assess the determination of dry weight, resulting in a total of 240 wells analyzed across the four measurements. To mitigate edge effects arising from temperature and evaporation, the 36 wells around the edge of the plate were excluded from analysis and loaded with 200 µL of blank media [28]. The microplates were incubated at 23 ± 0.5 °C, and the OD measured at 405 nm at 20, 40, 60, and 80 h post-inoculation using a Synergy^TM^ HT MultiDetection microplate reader with Gen5 software. At each time point, one microplate per isolate was collected after OD measurement and stored at −20 °C. The OD values of all of the microplates were also measured after conidia settled (15 min post-inoculation) to establish the baseline reading OD of each culture. This baseline reading was subtracted from subsequent measurements to determine changes in OD for the construction of standard curves.

To quantify the changes in biomass of the microcultures, mycelial mats were extracted for the determination of dry weight. Thawed microplates were centrifuged in an Eppendorf Centrifuge 5810R (1968× *g*) at 4000 rpm for three minutes to force fungal material to the bottom of the wells, and the remaining media supernatant was removed using a pipette. The wells were then filled with 200 µL of 99% ethanol, and mycelia were scraped from the bottom of the well with a pipette tip to re-suspend the fungal material. The entire content of the well was then transferred to a pre-weighed aluminum weigh boat using a cut pipette tip. The process was repeated three times, refilling each well with 200 µL of ethanol and scraping to ensure complete removal of residual mycelial matter and rinsing of the pipette tip. Thus, a total of 600 µL of ethanol was utilized to thoroughly wash each well. Complete extraction of mycelial material from the wells was confirmed through microscopic examination of the microplates. If any residual mycelial material was observed, the extraction process was repeated until the wells were free of any remaining matter. For each isolate, ten replicate wells containing resultant mycelial suspensions were pooled into pre-weighed aluminum boats, resulting in a total of six pooled dry weight measurements per microplate with an approximate volume of 6 mL. The pooled samples were subsequently dried in an oven for 72 h at 60 °C within a heat-resistant, lidded box. Finally, the dried samples were weighed on a precision scale (Sartorius ME36S Ultra Micro Balance, 31 g × 0.001 mg (Sartorius UK Ltd., Epsom, UK).

Standard curves were produced to establish the correlation between OD and change in biomass using the OD measurements and their corresponding pooled biomass samples. At each measurement time point, the base OD values were subtracted from the measured OD values, and the resulting OD values for the 10 pooled wells in the sample were averaged. Regression analysis was performed to construct standard curves of OD by dry weight using the R package stats [29]. To test for differences in relationships between the OD and dry weight between isolates, a pairwise comparison of slope estimates was conducted using lsmeans [30]. This analysis applies a *p* value multiplicity adjustment to the least squares means of each isolate using Tukey’s HSD. Linear regression was also performed to determine slope estimates for biomass added over time for each isolate. All statistical calculations were conducted in R Version 1.4.1717 [29].

### 2.4. Comparison of OD Growth Measure to Radial Growth Measure on Agar Plates

To evaluate how the microspectrophotometric method compared to traditional growth assays performed on solid media, the growth of two isolates (*M. guizhouense* KVL 19_28 and *M. acridum* KVL 04_55) was measured using both microspectrophotometry and radial expansion methods. These experiments were conducted under the same conditions at 23 °C in SDY/4 and SDAY/4 media, respectively. For the microspectrophotmetric analysis, media and conidial suspensions were prepared as described in Section 2.2 in a microplate (n = 96). OD measurements were taken during the linear phase of the logistic growth curve (24 h post-inoculation) at 405 nm in 8-hour intervals. This provided a total of five measurements. To estimate the biomass, the linear equations deduced from the standard curves were applied to the spectrophotometric measurements. This allowed for the determination of the biomass estimation based on the spectrophotometric data.

To conduct the radial expansion analysis, for each isolate, four Petri dishes (90 mm × 15 mm, vented) of SDAY/4 were each inoculated with five cultures (n = 20). The inoculation was performed using 5 µL of a conidial suspension with a concentration of 2.4 × 10^4^ spores per mL, prepared according to the methods described in Section 2.1. Germination of the conidial suspension was confirmed as >99% after 24 h on a plate of SDAY/4, following the methods detailed in Section 2.1. The growth was recorded every 4 days, starting from the first day of detectable mycelium formation (day 4) until day 11. To calculate the radial expansion, the plates were digitally photographed at each measurement time point, and the colony area was calculated in ImageJ Version 1.53s [31].

## 3. Results

### 3.1. Growth Curves and Correlation of Biomass with OD

First, it was determined that the entire growth curve of a *Metarhizium* fungus (isolate KVL 12_37, *M. brunneum*) could be captured within a short time period (96 h) using microspectrophotometry (Figure 1A). The distinct phases of fungal growth, namely the lag, log (exponential), and stationary phases, were clearly discernible in the preliminary growth curves obtained during the pilot study (Figure A1). Second, standard curves were produced to quantify the change in culture OD with a corresponding increase in dry weight of fungal biomass over the linear growth phase of the logistic growth curve for two isolates each of *M. guizhouense* (Figure 1B), *M. acridum* (Figure 1C) and *M. brunneum* (Figure 1D). The linear growth phase was determined between 20 and 80 h after inoculation, based on the specified parameters for media and concentration of conidial suspension (Figure 1A). The resulting standard curves for dry weight and OD showed high correlation coefficients across all isolates (R^2^ = 0.93–0.95) (Figure 1), but with varying biomass incorporation (i.e., slopes) between isolates (Figure 2). In the pairwise comparison of the slope estimates for different isolates, some slope estimates for the relationship between OD and dry weight significantly differed from others (Figure 2 and Table 1). Notably, the *M. guizhouense* isolates (KVL 19_28 and KVL 19_24) exhibited distinct slope estimates compared to most other isolates, as well as each other (Table 1). The slope for isolate KVL 19_28 differed from all other isolates, with significantly larger dry weight estimation compared to all other isolates, with the exception of KVL 16_36 (*M. brunneum*; Table 1). Isolate KVL 19_24 was significantly different from KVL 18_06 (*M. acridum*) and KVL 16_36 (*p* < 0.05), with a smaller dry weight estimation compared to these two isolates (Figure A2).

The biomass accumulated over time was determined for each isolate using slope coefficients extracted from the linear models (Figure 3A), and differences in the growth rates among isolates were clearly distinguishable between some isolates (Figure 3B). Isolates KVL 19_24 (*M. guizhouense*) grew more slowly, and had a lower overall biomass than all other isolates (Figure 2B and Figure A2). In contrast, isolate KVL 18_06 of *M. acridum* displayed more rapid growth, and achieved a higher overall biomass compared to the other isolates (Figure 2B and Figure A2).

### 3.2. Comparison to Traditional Method and Proof of Concept

We applied this method to measure the growth rates of two species of *Metarhizium* (*M. guizhouense* KVL 19_28 and *M. acridum* KVL 04_55) over 56 h, from 20 to 80 h after inoculation (Figure 4A), in conjunction with performing the same growth analysis using a radial expansion assay over 11 days (Figure 4B). In the microspectrophotometric analysis, the respective linear equations for the two isolates derived from the standard curves were applied to the measured OD values to predict the dry weight per well. End point values between both methods showed similar results: the KVL 04_55 (*M. acridum*) isolate had a faster growth rate, growing more overall compared to KVL 19_28 (*M. guizhouense*), whether that was on the solid agar medium (Figure 4A) or using the new technique in the liquid medium (Figure 4B).

It is important to note that Figure 4A represents a shorter time period of 9 days, whereas Figure 4B covers the first 56 h only (approximately 2.33 days). This distinction underscores the significance of our method, as it reveals detailed differences in growth rates that would not be apparent otherwise. While the end points in both figures show similar results, the microspectrophotometric method provides a more detailed analysis, exposing variations in the initial growth rate. For instance, the KVL 04_55 isolate exhibits slower initial growth, only overtaking the KVL 19_28 isolate after the 40-hour sampling time point (Figure 4B). This level of detail allows us to discern growth rate differences that would not be evident from the 1–3 day growth analysis on solid media. Furthermore, an important aspect of the new assay (Figure 4A) is the low measurement error observed over this short time period. This reduced error enhances the reliability and accuracy of the growth rate measurements obtained. Finally, we developed linear models to assess the relationship between time and growth for the two measurement methods. Despite the inability to directly compare the regression lines due to the utilization of different measurement methods, the relationship between time and growth appears similar for the different isolates. The remarkable similarity observed in the regression lines (Figure 4C,D) indicates that the slopes of the growth rates for the two species of *Metarhizium* are quite similar between assays, although they are at different time scales. This similarity suggests that, regardless of potential variations in initial growth rates or growth patterns observed between the two assays (Figure 4A,B), the overall growth rates for both species exhibit a comparable trajectory over time in the context of both measurement methods.

Continuous measurement of OD over time allowed in situ fungal growth to be monitored at a fine scale and with low measurement error, which demonstrated this as a feasible technique for the two isolates of *M. brunneum* (Figure 1D and Figure A1). This method can also be used to measure ODs at greater intervals, or for end point determination of growth (Figure 4). The growth curve generated for *M. brunneum* KVL 12_37 and KVL 12_30 clearly differentiates the lag, exponential, and stationary phases of the growth curve from continuous readings (Figure 1A and Figure A1).

## 4. Discussion

In this study, we demonstrate that microspectrophotometry can be used to capture the growth curve of entomopathogenic fungi in situ, and that the change in absorbance can be directly correlated with an increase in fungal biomass. Optical density (OD) values correlated strongly with biomass, and it was demonstrated that it is possible to produce a standard curve for the quantification of dry weight from OD by extracting, drying, and weighing the microcultures of mycelia on a precision scale (Figure 1).

In addition, we showed that change in the OD accurately represents biomass accumulation. This is not always the case in measurements of radial expansion, as the expansion of fungal colonies on solid media can often be equal in area, while having significantly different densities of mycelial growth [20]. Fungal cultures grow in multiple planar dimensions on agar (i.e., upwards and downwards, as well as across the substrate surface), and the hyphal mass can thus be more or less dense depending on nutrient availability [32]. The microspectrophotometric technique provides a quantitative measurement of hyphal density that is more uniformly distributed throughout the microculture, which produces a better representation of three-dimensional hyphal density than measuring two dimensions. This technique also makes it possible to capture specific effects from environmental changes on different fungal growth phase features that are only detectable at small time scales (Figure 4B), which produce atypical growth curves, e.g., rapid depletion of a primary nutrient source followed by a secondary phase of logarithmic growth [33]. Furthermore, the solid medium does not accurately represent the environment which entomopathogenic fungi encounter during infection (i.e., insect hemolymph). Given that most entomopathogenic fungi are filamentous with similar ecologies (i.e., penetration of host cuticle and subsequently spreading as individual cells through the insect hemocoel), this technique could be applied to investigate growth in other entomopathogenic species [34]. However, differences in biomass accumulation during growth as represented in the relationship between fungal dry weight and OD necessitates the production of standard curves specific to the isolate under investigation, although this would only need to be undertaken once per isolate to be applied to subsequent high throughput measurements.

In this study, we compared the use of microspectrophotometry with radial expansion analysis to assess the growth dynamics of two species of *Metarhizium, M. acridum* and *M. guizhouense*. Our findings demonstrated that the microspectrophotometric analysis yielded similar end point results and linear growth rates compared to the radial growth assay (Figure 4A–C). However, the microspectrophotometric technique provided the additional advantage of capturing differences in early growth rates that were not evident in the radial growth assay (Figure 4). The finer scale of measurement provided by the microspectrophotometric technique illustrates its value in accurately capturing different aspects of fungal growth curves that might be otherwise undetectable when measuring macroculture growth over many days. Compared to methods with fewer measurement points, this technique is better suited for detecting subtle differences in growth (Figure 4).

Importantly, our study revealed a different relationship between *M. acridum* and *M. guizhouense* through finer measurement intervals during critical growth phases, particularly the early linear phase, which could not be discerned through radial expansion measurements performed over several days (Figure 4). This development is significant for understanding the growth dynamics of fungi, as it has been previously shown that the classical growth curve does not always adequately describe the growth patterns of filamentous fungi [33]. Different aspects of the growth curve can change due to variables such as nutrition and the host insect environment. Capturing the growth pattern can offer important insights into various aspects of how fungi grow during infection. Notably, atypically shaped growth curves have been suggested to be the rule rather than the exception [33]. This arises from the depletion of distinct nutrients occurring at different rates, resulting in atypical-shaped growth curves, like bimodal growth peaks [33]. These curves reveal nutritive preferences that might be otherwise overlooked without the necessary sensitivity in measurements. Our study revealed that the *M. acridum* KVL 04_55 isolate generally exhibited more growth over time compared to other isolates (Figure 4). However, a notable finding was that in the early stages, this isolate displayed a significantly slower growth rate compared to the other isolate examined, *M. guizhouense* KVL 19_28. This observation provides important insights into the growth dynamics and pathogenic potential during early infection processes for this particular isolate [35].

Additionally, our findings have practical implications regarding the pathogenic potential of the different isolates and species measured. By comparing the growth rates and total biomass produced between isolates, our findings shed light on the potential speed of host invasion across different species of *Metarhizium*. Significantly differing slope estimates and total biomass production were observed not only between species, but also among isolates, indicating variations in pathogenicity potential within this in vitro setting (Figure 3 and Figure A2). This highlights the practical application of the method in assessing the virulence and pathogenic potential of the examined isolates and species.

Previous studies demonstrated the use of microspectrophotometry in monitoring the growth of filamentous fungi, but were unable to determine the dry weight of individual microcultures, and therefore relied on comparisons of indirect metrics to extrapolate the relationship between OD and fungal dry weight [10,13], or used microscopic measurements of hyphal extension to infer growth [10]. In this research, we inferred growth in microculture using mycelial dry weight, and generated standard curves for direct correlation. While the microspectrophotometric technique may provide limited phenotypic information compared to solid media bioassays, such as the measurement of spore production and colony color, it offers complementary advantages in capturing biomass build-up during the early phases of the growth curve, and obtains data rapidly. The speed at which growth data can be obtained using microspectrophotometry is a clear advantage, as linear growth of the tested *Metarhizium* isolates could be measured within a few days, whereas radial growth on agar plates typically takes 10–14 days.

The need to produce entomopathogenic fungi for biocontrol at massive scales makes it important to be able to investigate the effects of different media and nutrients on entomopathogenic fungal growth, and identify optimal growth parameters [36]. In addition, for experimental biologists, this method allows for large-scale experiments using growth as a primary measure of performance in areas of research such as fitness costs, adaptation [37], and niche quantification [38]. A more detailed picture of the different growth phases could provide an understanding of nutritional adaptation, for example, in revealing nutritive preferences [33].

In conclusion, this technique allows for the rapid generation of growth curves of entomopathogenic fungi at a fine timescale with many replicates, and in a medium that is more ecologically relevant to entomopathogenic fungi than what typical solid media bioassays provide. Furthermore, this approach has the potential for application to other species of filamentous entomopathogenic fungi under investigation, such as *Beauveria* spp., *Hirsutella* spp., *Cordyceps* spp., and *Lecanicillium* spp. [3]. The methodological developments described advance the applications of spectrophotometry to the monitoring of filamentous fungal growth in entomopathogenic fungi, and resolve the infeasibility of producing standard curves directly correlating change in OD with mycelial mass. This automated and high-throughput method for monitoring in situ fungal growth presented here will aid further studies on aspects affecting growth of these ecologically and commercially important organisms.

## Figures and Tables

**Figure 1 insects-14-00703-f001:**
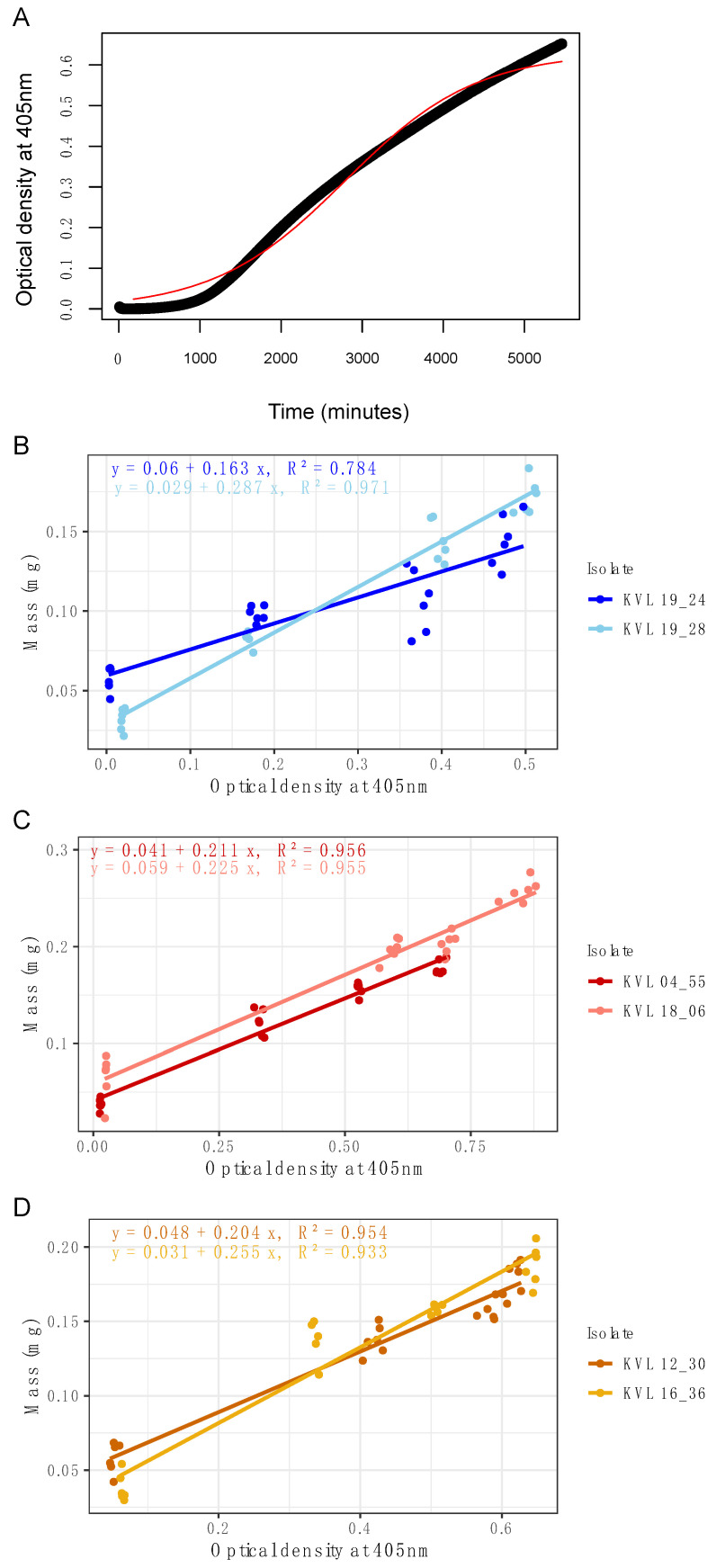
Standard curves and growth curve for *Metarhizium* spp. (**A**) Fitted growth curve over 96 h using continuous measurement (every 10 minutes) for *M. brunneum* KVL 12_30 for 96 averaged microcultures. The red line indicates fitted logistic growth model, and the black line is average absorbance readings for 96 wells at each time point. Standard curves for change in dry weight as a function of optical density (OD) for three species of *Metarhizium*: (**B**) *M. guizhouense* KVL 19_24 and KVL 19_28 (blue and light blue), (**C**) *M. acridum* KVL 04_55 and KVL 18_06 (red and light red) and (**D**) *M. brunnuem* KVL 12_30 and KVL 16_36 (dark yellow and yellow) cultured in a 96-well microplate. Shaded bands around regression lines indicate 95% confidence intervals.

**Figure 2 insects-14-00703-f002:**
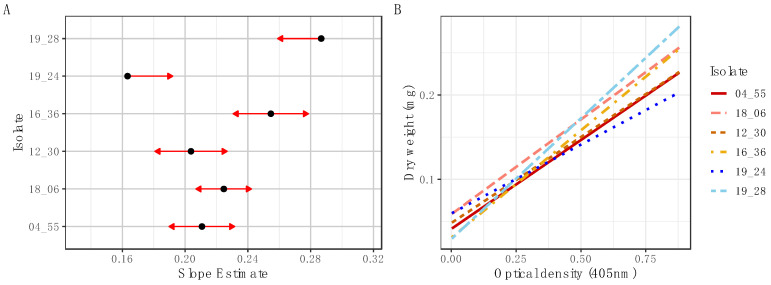
Comparison of slope estimates for dry weight as a function of change in optical density (OD) among six *Metarhizium* isolates. (**A**) Graphical comparisons of least squares means for each isolate’s slope estimate. Black dots indicate slope estimates for biomass as a function of change in OD. The shaded bands are corresponding confidence intervals at an alpha level of 0.1. Arrow lengths indicate the amount by which confidence intervals for differences cover the value 0. (**B**) Corresponding regression lines for dry weight as a function of change in OD for six *Metarhizium* isolates.

**Figure 3 insects-14-00703-f003:**
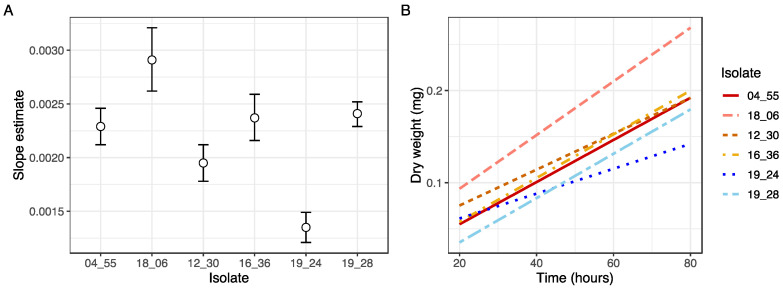
Biomass added over time for six *Metarhizium* isolates. (**A**) Slope estimates for linear phase growth rate of six isolates and three species of *Metarhizium* calculated using dry weight measured at 20-hour intervals from 20 to 80 h (milligrams of dry weight ~ time*isolate). Estimates are for ten pooled microcultures collected from 60 wells in a 96-well microplate, n = 6. White dots indicate the slope estimates, with SE bars calculated at an alpha level of 0.1. (**B**) Corresponding regression lines for dry weight (mg) as a function of time in hours in six *Metarhizium* isolates.

**Figure 4 insects-14-00703-f004:**
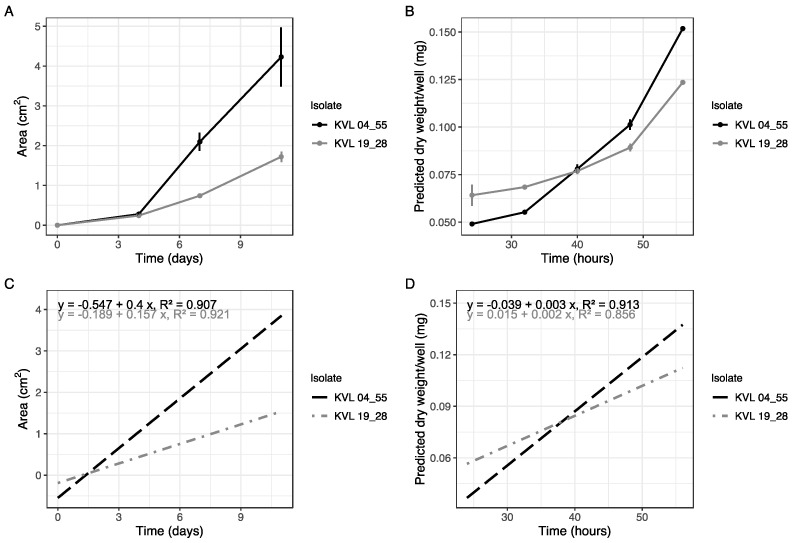
Growth curves for *Metarhizium acridum* KVL 04_55 and *M. guizhouense* KVL 19_28 produced on solid agar SDAY/4 media (n = 20) (**A**), using the new technique in SDY/4 liquid media (n = 96) (**B**), and regression lines for each (**C**,**D**). The equations derived from the standard curves for the relationship between absorbance and dry weight were applied to OD values measured from plates of *Metarhizium* grown over 56 h to deduce the biomass added over time for each isolate (**A**). Dots represent the mean increase in area/colony or predicted biomass/well at each time point, and bars indicate standard deviation. Regression lines (**C**,**D**) were obtained using the lm function with the method of least squares.

**Table 1 insects-14-00703-t001:** Pairwise comparison of slope estimates for dry weight as a function of change in OD for six *Metarhizium* isolates.

Contrast	Estimate	SE	*p* Value ^1^
04_55–12_30	0.0069	0.0167	0.998
04_55–16_36	−0.0439	0.0171	0.113
04_55–18_06	−0.0139	0.0142	0.924
04_55–19_24	0.0474	0.019	0.133
04_55–19_28	−0.076	0.0185	0.001
12_30–16_36	−0.0509	0.018	0.059
12_30–18_06	−0.0209	0.0153	0.746
12_30–19_24	0.0404	0.0198	0.324
12_30–19_28	−0.0829	0.0194	0.0005
16_36–18_06	0.03	0.0158	0.406
16_36–19_24	0.0913	0.0202	0.0002
16_36–19_28	−0.0321	0.0197	0.583
18_06–19_24	0.0613	0.0178	0.009
18_06–19_28	−0.062	0.0173	0.006
19_24–19_28	−0.123	0.0214	<0.0001

^1^ *p* values were adjusted for multiplicity using Tukey’s HSD.

## Data Availability

The data supporting the findings of this study are available upon reasonable request. Requests for access to the data can be directed to the corresponding authors and will be subject to approval by the data owners. Availability of the data is contingent upon compliance with relevant data use and privacy regulations.

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
