# Peer review of "A Rapid Method for Measuring In Vitro Growth in Entomopathogenic Fungi"

_insects, 2023, doi:10.3390/insects14080703_

Round 1

Reviewer 1 Report

The manuscript was about a rapid method for measuring in vitro growth in entomopathogenic fungi. It was generally well written and informative. This is an important topic and the method will reduce the time it takes to assess the growth of entomopathogenic fungi. It will also allow for a more efficient method to screen potential isolates for testing against pests based on their initial growth ability. There are some minor issues that need to be addressed prior to publishing. These are found below.

Although it’s a free format, I do believe that the references must be numbered in the body of the text and appear in order of appearance in the text and listed individually at the end of the manuscript.

Line 28 – Change ‘Here’ to “Herein”

Line 50 -52 should read like this “In biological studies, measurements of fungal growth over time on different defined media serve as a standard measure to assess performance and evaluate efficacy during isolation and testing…...” – The work fungi was used 3x in one sentence before.

Line 66-68 should be split to “Measurement of the radial expansion of fungal colonies on solid media is a widely used direct method to quantify growth. The method is straightforward and allows for the measurement of other phenotypic traits such as spore production and colony morphology (e.g. colour, branching pattern)….

Line 71 – Add a period after the citation and start a new sentence with “An”

Line 77 – Either combine with the above paragraph or start paragraph in a different manner form the use of “Furthermore,”. A new paragraph could be started with the sentence after.

Line 93 - Change ‘Here’ to “Herein”

Line 95 97 should be reworded/rephrased. It is confusing.

Line 104 – 106 – This sentence should be in discussion or conclusion.  – Change ‘holds the potential’ to ‘has the potential’.

In general lines 103 – 108 could be restructured. Or perhaps, just 103 -104. It demonstrates the method but for what? To save time? That’s more made clear in the last paragraph to tie it all together.

Lines 111-113 – Should be rephrased as “The growth of two isolates within each of three different species of Metarhizium were compared to investigate variation among species and isolates. This was assessed in addition to the effect on the relationship between dry weight and OD.” …. or something as such.

Why was Metarhizium selected instead of Beauveria or another? Perhaps a sentence that identifies that and explains why would be beneficial in the intro. “Metarhizium (Metschnikoff) Sorokin (Order Hypocreales: Family Clavicipitaceae) was selected because…..”

Manufactures and associated locations of primary items like Petri dishes, plates, agar etc. should be included.

Please check use of multiplication sign vs alphabetical x. Examples – line 132 1000x and line 135 between coverslip dimensions and elsewhere.

Line 150 – Delete “and furthermore” and start sentence with “Pilot”

Line 156 – What 4 timepoints?

Its not entirely clear how many replicates for each isolate were tested. Was it 96 x 4? Should be clarified.

Line 226 – Start sentence as “First, it was determines that…”

Line 230 – Change ‘secondly’ to “Second,”

Line 289 – delete ‘both methods indicated’ after the :

Line 327 – add a ‘the’ before ‘two isolates’

There is a formatting issue with the x axis in figure 1. The text is strange and squished together, especially with the ‘i’s. This should be fixed is possible. Same for other figures.

Lines 376 – 381 are redundant. Should be rephrased.

Lines 382 – 388 also need to be reworked to improve flow.

Line 392 – Add “like” before “M.”

Line 395 – Needs a citation to back that up.

Line 410 – Should read “Herein, we inferred growth in microculture using mycelial dry weight and generated standard curves for direct correlation.”

Reviewer 2 Report

Comments to Author:

Ms. Ref. No.: [Insects] Manuscript ID: insects-2527780

Title: A rapid method for measuring in vitro growth in entomopathogenic fungi

Overview and general recommendation:

The objective of this study was to develop a method for measuring the growth of entomopathogenic fungi assessing in vitro growth through spectrophotometry.

I disagree that is a new method or a novel technique as mentioned in the abstract. This technique is currently used in laboratories to assess fungal growth. As well as, there are many papers related to this subject, some of them:

1) Paisley D. and others, Correlation between in vitro growth rate and in vivo virulence in Aspergillus fumigatus, Medical Mycology, Volume 43, Issue 5, August 2005, Pages 397–401, https://doi.org/10.1080/13693780400005866

2) Banerjee, U.C., Chisti, Y. & Moo-Young, M. Spectrophotometric determination of mycelial biomass. Biotechnol Tech 7, 313–316 (1993). https://doi.org/10.1007/BF00150905

3) In the references of the ms: Granade TC, Hehmann MF, Artis WM. Monitoring of filamentous fungal growth by in situ microspectrophotometry, fragmented mycelium absorbance density, and 14C incorporation: alternatives to mycelial dry weight. Appl Environ Microbiol. 1985 Jan;49(1):101-8. doi: 10.1128/aem.49.1.101-108.1985.

Minor suggestions are:

Lines 16 and 28: Novel? Technique. This approach is not new. The novelty consists in applying this method on Metarhizium spp.

Lines 32 and 35: Metarhizium instead of Metahrizium

Line 106: Cordyceps instead of Isaria. Shah & Pell (2003) did not mention Isaria. They mentioned Paecilomyces fumosoroseus. Today referred to as Cordyceps fumosorosea.

Line 215: Petri dishes instead of petri dishes

Figure 1 (D): What is the measurement on the y-axis? Absorbance? What is the measurement on the x-axis? Minutes?

Line 292: new? technique or just technique

Line 296: new? Technique

Line 312: new? assay

Line 402: This is not a new method

Check out italics in scientific names.

On the one hand, I found the paper to be overall well written and the methodology very well described. The current study is on a topic of relevance and general interest to readers who work with entomopathogenic fungi.

Reviewer 3 Report

This paper is well-written and offers a new method for quantifying the growth of entomopathogenic fungi and will be applicable to other related fungi. The methods are super detailed and the results are nicely presented.

Reviewer 4 Report

This manuscript nicely describes a new technique based on spectrophotometry to assess, in vitro, the growth of EPF. The description of the methods is clear and the supporting information is well presented, with nice illustrative figures and analysis that detail the suggested methods for future use in Labs. I have a very few minor comments with respect to the result section that I have outlined below.

Other than those, I would like to add that it is nice to read this work which will facilitate handling EPF. This work contributes significantly to that knowledge.

For the result section:

- I found it a bit confusing to talk about Figure 1D before the section related to "standard curve sections and figures". I suggest the authors to please pay attention to presenting the text following the order of the figures. 

- Please write the units for the Time in Figure 1. D
